# Authentication of Medical Staff with Protective Gear-Wearing: Utilization of Handwritten Letter Characteristics and Machine Learning

Kyoka Shirae and Chinthaka Premachandra

*Abstract*— In recent years, various human identification technologies, such as facial recognition, have been increasingly adopted for security purposes. However, these conventional biometric systems face significant challenges in scenarios where individuals are required to wear protective clothing, which can obscure facial features and fingerprints. This paper introduces an innovative approach designed to overcome such obstacles by focusing on the individuality of handwritten characters, offering a viable alternative when traditional biometric identifiers are unusable.

Our study proposes and evaluates a method that uses machine learning to analyze and learn the unique features of handwritten characters. This approach is independent of typical biometric traits such as facial features and fingerprints, thus providing a novel solution for identity verification in specialized environments like laboratories or hazardous material handling areas where protective gear is mandatory.

We developed a model using a random forest algorithm trained on binary images of distinct handwritten characters written by participants. The selection of handwritten characters as the basis for our study stems from their inherent uniqueness to each individual, similar to other biometric markers. The training process involved extracting and learning the subtle differences in handwriting styles, strokes, and patterns that are difficult to replicate or disguise.

The effectiveness of this methodology was validated through rigorous testing. The random forest model was applied to a new set of data to determine its accuracy in identifying the correct writer of the handwritten samples. Impressively, the model achieved a correct identification rate of 97.8%, underscoring the potential of handwriting-based identification as a robust and reliable security measure.

## I. INTRODUCTION

In recent years, various human identification technologies have been deployed as security measures. Common examples include fingerprint, facial, and iris recognition, which are utilized in smartphones and homes. However, the spread of the novel coronavirus has prompted the recommendation of masks and protective gear in various workplaces for safety reasons. In such contexts, face and fingerprint recognition can be challenging. Traditional face orientation estimation techniques rely on detecting distinct facial features such as the nose, eyes, and mouth, and estimating face orientation based on the movement of these features [1][2][3][4][5]. However, these methods struggle when the face is turned sideways or covered by a mask, as key features become obscured, rendering the estimation of face direction impossible.

For this reason, studies have explored effective identification methods that remain functional even when protective clothing is worn. One approach involves attaching AR markers to the surface of protective gear [1]. This paper introduces a novel technology for identifying individuals in protective clothing using AR markers, which does not rely on visible facial or fingerprint features. This technology represents a shift towards non-appearance-based identification methods.

Writer identification has also been extensively researched and can be categorized into two main types. The first category includes methods that identify individuals based on images of handwritten characters using neural networks [2][3][4][5][6][7], focusing on features such as the line patterns of characters. These methods are generally sensitive to image distortions and rotations. The second category involves dynamic characteristics like writing pressure and speed, as demonstrated in the prototype of a high-sensitivity pressure pen designed for fast writers and subsequent authentication experiments [8][9][10][11][12]. Recognition accuracy in these methods generally exceeds 90%. However, their reliance on specialized pens limits their versatility.

Consequently, this paper explores identifying writers using only images of handwritten characters. We employ machine learning to analyze the distinct characteristics of handwritten characters, using this data to recognize individuals. We collected 1,100 images of single characters from five different character types, created by four writers. These images were binarized to expedite processing and then trained using machine learning techniques. Random forests were chosen for training due to their high predictive accuracy, rapid training and identification capabilities for large datasets, and transparent results.

Figure 1 displays images of the alphabet letter "M" written by four individuals. As the shape of the letter varies from person to person, we leverage this uniqueness to determine the writership. In this paper, we describe a novel preprocessing and annotation process applied to each character before training for writer classification. The results indicate that our proposed method can accurately identify the writer with a 97.8% success rate.

Figure 1. "M" by four writers.

## II. OVERVIEW OF THE PROPOSED SYSTEM

### A. Overall flow of the proposed writer identification method

The overall flow of the proposed writer identification method is illustrated in Fig. 2. Initially, after the images are binarized, a square-cropped image is prepared with the handwritten characters centered. This binary image serves as the basis for identifying the writer using machine learning. The binarization process is designed to accentuate the shape features of the characters, which are crucial for identification. In this study, we utilize a random forest, an ensemble learning method that employs multiple decision trees. Random forests are effective for both classification and regression tasks.

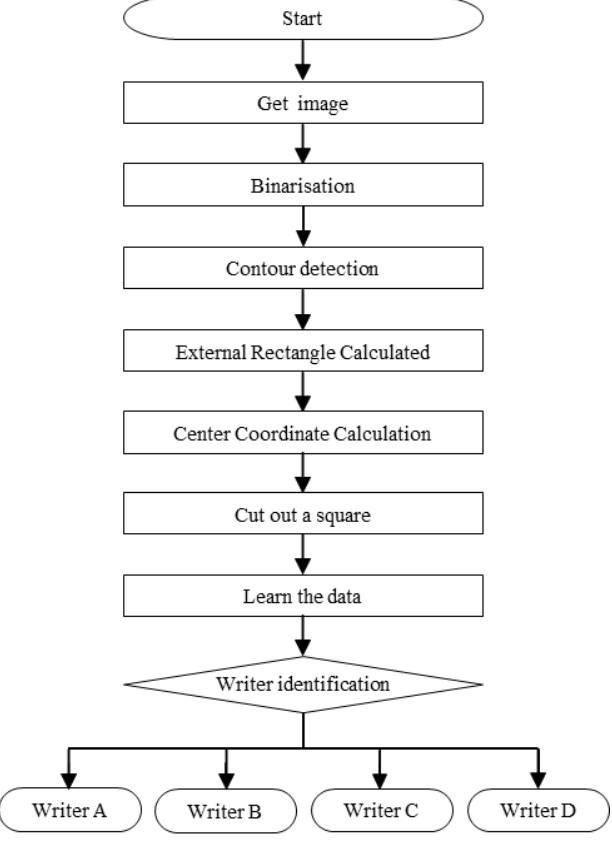

Figure 2. Overall flow of the proposed writer identification method

## III. MACHINE LEARNING FOR WRITER IDENTIFICATION OF HANDWRITTEN TEXT IMAGES

### A. Data Set Preparation

In this study, image data were scanned and collected using a Kyocera Taskalfa 5052ci color multifunction printer. We focused on handwritten images of five alphabetic characters: "G," "M," "Q," "W," and "Z." A total of 22,000 images were gathered, with four writers contributing 1,100 instances of each character type. Of these, 800 images were used for training, 200 for validation, and 100 for testing. To ensure consistency across samples, the same ballpoint pen was used for all writings. Additionally, to minimize variations in letter size and prevent excessive slanting, writers wrote within 100 pre-drawn squares on a sheet of A4 paper.

The process for creating binarized images is detailed as follows: Initially, scanned images are acquired from the multifunction device and then binarized to optimize learning and inference times. Figure 3 presents examples of images before and after the binarization process.

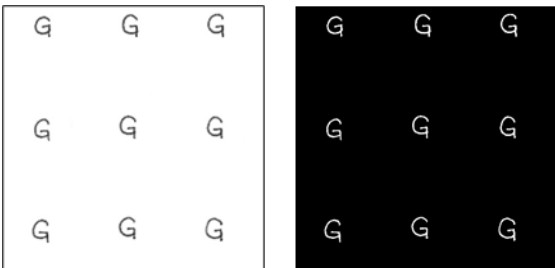

Figure 3. Example images before and after binarization

The contour detection process identifies 100 characters and calculates the bounding rectangle for each detected character. Upon calculation, four values are returned: the x-coordinate (x), y-coordinate (y), width (w), and height (h) of the upper left corner of the rectangle. These dimensions are illustrated in Fig. 4. From these four values, the center coordinates ($cx, cy$) of the character are determined. The center coordinates are computed using the following equation (1):

$$\begin{cases} cx = x + \dfrac{w}{2} \\ cy = y + \dfrac{h}{2} \end{cases} \qquad (1)$$

Then, as demonstrated in Fig. 5, the dataset is cropped into a square using the center coordinates previously calculated. The dataset was prepared following this method, ensuring that each character is centered within its respective image frame.

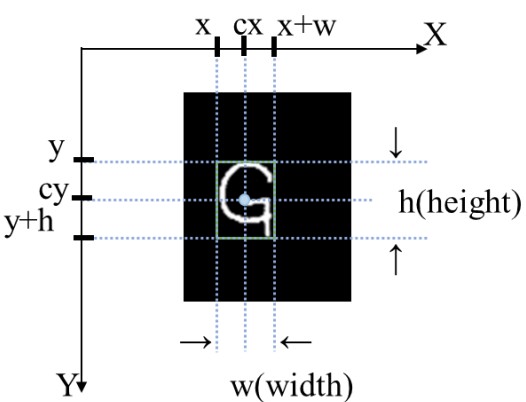

Figure 4. Calculation of center coordinates by bounding rectangle

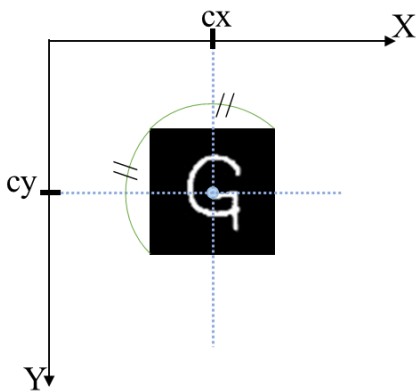

Figure 5. Example of image cropping

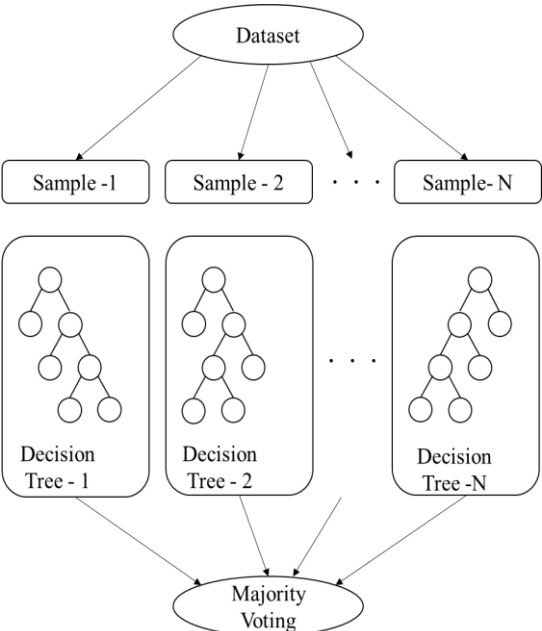

Figure 6. Conceptual Diagram of Random Forest

## B. Random Forest

In this section, we describe random forests. In this study, we utilized a random forest, and a conceptual diagram of this method is shown in Fig. 6.

Random forest is a machine learning algorithm classified under ensemble learning as bagging. It performs class classification by taking a majority vote from the results of multiple decision trees constructed from randomly selected training data and explanatory variables. The user must manually set the number of trees and their depth; these settings are detailed in section 3.3 under Grid Search.

It is well known that decision trees, when used alone, may not be highly accurate and are susceptible to overfitting. Ensemble learning addresses these disadvantages by using multiple weak learners to achieve highly accurate predictions [14]. Random forest operates by performing classification based on the results of multiple decision trees,

each created from different subsets of the original dataset, allowing for feature overlap. This process ensures that each decision tree is slightly unique. Random forests have demonstrated high performance in both classification and regression tasks, making them a popular choice in the field of image recognition.

There are three main reasons for using random forests in this study:

**High Prediction Accuracy:** The primary goal of this research is to identify individuals wearing protective clothing to enhance security measures. Thus, high prediction accuracy is crucial.

**Rapid Learning and Identification with Large Datasets:** Our experiments involved four writers and five character types. In practical applications, the number of writers and character types, as well as the volume of data, are expected to increase significantly. Random forests enable fast learning and identification, which facilitates smooth recognition even under these conditions.

**Transparency of Results:** Random forests allow for the calculation of feature importance and the visualization of tree structures, making it clear how classifications are determined. This transparency is beneficial for improving the system's accuracy and reliability.

For these reasons, random forests were selected for use in this study.

## C. Grid search

Hyperparameters are parameters that control the behavior of an algorithm. In the case of random forests, critical hyperparameters include the number of trees and the depth of these trees. Tuning these parameters is crucial for developing a highly accurate model. Grid search is a commonly used method for identifying the optimal parameters. This approach involves specifying the values for each parameter (for instance, the number of trees to generate) in an array and systematically evaluating all possible combinations in a round-robin fashion. While this method can enhance model accuracy, it is time-consuming due to its exhaustive nature.

## IV. VERIFICATION EXPERIMENT

### A. Experimental Results

In this chapter, we present the results of validating the personal identification model developed using the previously described learning methods. Figure 7 displays the confusion matrix generated from the test data. In this matrix, the vertical columns represent the correct labels, while the horizontal columns indicate the predicted labels. Each cell in the matrix denotes the number of samples correctly predicted for each label. The confusion matrix reveals that four misclassifications significantly impact overall accuracy: data from writer A is often mis-predicted as writer C; data from writer B is frequently mis-predicted as writers A and C; and data from writer C is commonly mis-predicted as writer A. Conversely, predictions for writer D are relatively accurate.

| | Predicted class | | | |
|---|---|---|---|---|
| | A | B | C | D |
| A | 489 | 4 | 7 | 0 |
| B | 8 | 480 | 10 | 2 |
| C | 8 | 4 | 488 | 0 |
| D | 0 | 2 | 0 | 498 |

Figure 7. Confusion matrix (4-class classification by writer)

We explain the elements of the confusion matrix used for two-class classification, with each element represented as a true positive (TP), true negative (TN), false positive (FP), and false negative (FN). Figure 8 illustrates this matrix configuration.

**TP**: This is the count of correctly predicted instances where the actual writer is identified accurately.

**TN**: This represents the number of correct predictions where a different writer is correctly identified as not being the actual writer.

**FN**: This occurs when the actual writer is incorrectly predicted as not being the writer.

**FP**: This is the count of incorrect predictions where a different writer is mistakenly identified as the actual writer.

From these values, we derive the metrics of Accuracy, Precision, and Recall.

Figures 9-12 depict a two-class confusion matrix derived from Fig. 8, where one class is designated as Positive and the other as Negative. These figures are consolidated and presented again in Fig. 13 for comprehensive visualization.

| | Predicted class | |
|---|---|---|
| | Positive | Negative |
| Positive | TP | FN |
| Negative | FP | TN |

Figure 8. Confusion matrix (each element)

| | Predicted class | |
|---|---|---|
| | A | Others |
| A | 489 | 11 |
| Others | 16 | 1484 |

Figure 9. Confusion matrix (Positive=A)

| | Predicted class | |
|---|---|---|
| | B | Others |
| B | 480 | 20 |
| Others | 10 | 1490 |

Figure 10. Confusion matrix (Positive=B)

| | Predicted class | |
|---|---|---|
| | C | Others |
| C | 488 | 12 |
| Others | 17 | 1483 |

Figure 11. Confusion matrix (Positive=C)

| | Predicted class | |
|---|---|---|
| | D | Others |
| D | 498 | 2 |
| Others | 2 | 1498 |

Figure 12. Confusion matrix (Positive=D)

| | Elements | | | |
|---|---|---|---|---|
| | TP | TN | FP | FN |
| A | 489 | 1484 | 16 | 11 |
| B | 480 | 1490 | 10 | 20 |
| C | 488 | 1483 | 17 | 12 |
| D | 498 | 1498 | 2 | 2 |

Figure 13. Summary of each element in Fig. 9-12

Based on the above, we first discuss the percentage of correct answers. The correct response rate is the percentage of correct responses to all predictions, and is calculated by the following equation (2).

$$Accuracy = \frac{TP + TN}{TP + FP + FN + TN} \quad (2)$$

The overall percentage of correct answers is calculated differently from the above formula for the four classes. It is calculated by dividing the number of correct answers by the total number of predictions.

Next, the goodness-of-fit rate is the percentage of data that is actually positive among those predicted to be positive, and is calculated by the following equation (3).

$$Precision = \frac{TP}{TP + FP} \quad (3)$$

Finally, the reproducibility is the percentage of those that are predicted to be positive among those that are actually positive, and is calculated by the following equation (4).

$$Recall = \frac{TP}{TP + FN} \quad (4)$$

After all the classes are set as Positive and the evaluation values are all available, the overall fit rate and the reproducibility rate are also calculated by taking the macro average. When the overall fit rate is PreA, the fit rate of class A is PreA, and similarly PreB, PreC, and PreD, the following equation (5) is used to obtain the overall fit rate.

$$Pre_{all} = \frac{Pre_A + Pre_B + Pre_C + Pre_D}{4} \quad (5)$$

The macro average is the average of the values calculated for each class. The reproducibility is calculated in the same way. Table 1 summarizes the results of the correctness rate, conformance rate, and reproducibility rate.

Table 1  shows a summary of these three data.

| Writer | A | B | C | D | Over all |
|---|---|---|---|---|---|
| Accuracy[%] | 98.8 | 98.7 | 98.6 | 99.9 | 97.8 |
| Precision[%] | 96.8 | 98.0 | 96.6 | 99.6 | 97.8 |
| Recall[%] | 97.8 | 96.0 | 97.6 | 99.6 | 97.8 |

A sample of the results of the predictions made by randomly selecting data for the test data is shown in Fig. 14.



Figure 14.  Result Samples

In Figure 14, the labels 'A', 'B', 'C', and 'D' at the bottom of each image represent the correct author on the left and the predicted author on the right. The color coding indicates the accuracy of the prediction: blue signifies that the correct label and the predicted label match, whereas red denotes a discrepancy between them.

Data instances where the correct and predicted labels differed were specifically analyzed to examine the breakdown by character type. The results of this analysis are presented in Table 2. The characters "G", "M", "Q", "W", and "Z" were the ones selected for the authors to write in this study.

Table 2  Misclassified Character Type Breakdown

|  | G | M | Q | W | Z | Total |
|---|---|---|---|---|---|---|
| A | 3 | 1 | 5 | 1 | 1 | 11 |
| B | 14 | 1 | 0 | 0 | 5 | 20 |
| C | 4 | 0 | 2 | 1 | 5 | 12 |
| D | 0 | 0 | 0 | 2 | 0 | 2 |
| Total | 21 | 2 | 7 | 4 | 11 | 45 |

The confusion matrix of character types is shown in Fig. 15-19.

| | | Predicted class | | | |
|---|---|---|---|---|---|
| | | A | B | C | D |
| True class | A | 97 | 2 | 1 | 0 |
| | B | 0 | 86 | 5 | 1 |
| | C | 3 | 1 | 96 | 0 |
| | D | 0 | 0 | 0 | 100 |

Figure 15.  Confusion matrix (Character type "G")

| | | Predicted class | | | |
|---|---|---|---|---|---|
| | | A | B | C | D |
| True class | A | 99 | 1 | 0 | 0 |
| | B | 0 | 99 | 0 | 1 |
| | C | 0 | 0 | 100 | 0 |
| | D | 0 | 0 | 0 | 100 |

Figure 16.  Confusion matrix (Character type "M")

| | | Predicted class | | | |
|---|---|---|---|---|---|
| | | A | B | C | D |
| True class | A | 95 | 0 | 5 | 0 |
| | B | 0 | 100 | 0 | 0 |
| | C | 2 | 0 | 98 | 0 |
| | D | 0 | 0 | 0 | 100 |

Figure 17.  Confusion matrix (Character type "Q")

| | | Predicted class | | | |
|---|---|---|---|---|---|
| | | A | B | C | D |
| True class | A | 99 | 1 | 0 | 0 |
| | B | 0 | 100 | 0 | 0 |
| | C | 0 | 1 | 99 | 0 |
| | D | 0 | 2 | 0 | 98 |

Figure 18.  Confusion matrix (Character type "W")

| | | Predicted class | | | |
|---|---|---|---|---|---|
| | | A | B | C | D |
| True class | A | 99 | 1 | 0 | 0 |
| | B | 0 | 95 | 5 | 0 |
| | C | 3 | 2 | 95 | 0 |
| | D | 0 | 0 | 0 | 100 |

Figure 19.  Confusion matrix (Character type "Z")

### B.  Discussion

In this study, using the Random Forest classifier—a machine learning method—we successfully identified the writer from binary images of handwritten single letters with a high probability, achieving a correct prediction rate of 97.8%. The sample of estimation results for the test data shown in Fig. 15 confirms the accuracy of these predictions. These findings underscore the effectiveness of the proposed method.

However, as indicated in Fig. 7, while writer D was almost always correctly identified, several misclassifications occurred for other writers. We aim to investigate the causes of these inaccuracies and explore potential solutions.

As detailed in Table 2, the accuracy of predictions varies significantly by letter type. Characters such as G, Z, Q, W, and M are the most prone to misclassification, in that order. Our analysis of incorrectly predicted images suggests that more complex characters, which typically involve more strokes, are more likely to be misidentified. These misshaped characters diverge from the typical forms expected of the writer, even though they were written by the same individual. Given that our data set includes only four writers and five character types, the sample size may be too small to conclusively determine trends. Expanding the experiment to include more handwriting styles and writers would likely provide more definitive insights.

The confusion matrices from Fig. 15-19 show that there were eight instances with more than two misclassifications, and 75% of these errors were biased (defined here as a difference of two or more samples). This suggests a similarity in character writing among certain writers. Notably, over half of the misclassifications involved data for the correct answer label B being classified as C, and vice versa. This indicates that a simultaneous classification of not only the writer's identity but also the character type might enhance the model's accuracy in capturing the individuality of each writer's style.

In the case of the four writers studied, distinct handwriting characteristics recognizable by the human eye were noted. This observation raises concerns about the model's ability to accurately classify handwriting from individuals attempting to mimic the four studied writers, warranting further verification.

Finally, we discuss the practical application of the system. If the results from single character identifications are aggregated and the writer of a string of characters is identified by majority vote, the accuracy could be sufficient for security purposes. However, the large dataset used in this study—1,100 samples per character type—places a significant burden on computational resources. Going forward, it will be crucial to find a balance between the volume of training data and the model's accuracy, aiming to reduce data requirements while maintaining robust performance. Increasing the amount of training data could also be considered to enhance system effectiveness..

## V.  Conclusions

In this study, we explored a machine learning method for identifying writers based on the individuality of handwritten characters, aimed at personal identification when wearing protective clothing. We created a dataset of 22,000 square, binary images of uniform size with the handwritten characters centered. Using Random Forest, an ensemble learning algorithm, we classified and identified writers across classes. The results demonstrated high accuracy in writer identification, confirming the effectiveness of using machine learning to recognize the individuality of handwritten characters.

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
