# OpenReview forum: "Authentication of Medical Staff with Protective Gear-Wearing: Utilization of Handwritten Letter Characteristics and Machine Learning"
_IEEE.org/ICIST/2024/Conference — IEEE ICIST 2024 Conference Submission_

### Official Review · Reviewer_euB3 · 2024-08-21
**accept**

**Rating:** 7
**Confidence:** 3

**Review:**

This paper considered a machine learning method for identifying writers based on the individuality of handwritten characters, aimed at personal identification when wearing protective clothing. The theory is correct and can be accepted after responding the following comments.
(1)	In the introduction, it is not enough to state the current work. It should be expended and reconstructed.
(2)	There are many typos and grammar errors. The authors should have a native English speaker or software packages to perform the editing check.
(3)	Please check if you need to update your Introduction/Related Work section to include latest closely relevant references that have appeared in journals and/or conferences in the past two years.

---

### Official Review · Reviewer_XMZk · 2024-08-21
**accept**

**Rating:** 7
**Confidence:** 3

**Review:**

Comment: This study proposes and evaluates a method that uses machine learning to analyze and learn the unique features of handwritten characters. It provides a feasible alternative for the situation that the traditional biometric identifiers are unusable. The theory is correct and can be accepted after responding the following comments.
(1) More comprehensive literature review is needed to clarify the research gap and research motivation.
(2) The format of several paragraphs in the article is inaccurate, and there is no first line indentation.
(3) In the end of the conclusions, some research directions are suggested to be added.

---

### Official Review · Reviewer_xVzA · 2024-08-22
**This article is very interesting and a good one**

**Rating:** 7
**Confidence:** 3

**Review:**

This paper proposed and evaluated a method that uses machine learning to analyze and learn the unique features of handwritten characters. The obtained result is valuable and can be accepted if the following problems can be clarified.
(1) In the introduction, the shortages of those relevant studies are suggested to be further summarized.
(2) In the end of Section 1, the organization of this study is suggested to be summarized.
(3) There exist several spelling and grammar errors. Please check carefully and further polish
(4) The future work is missing in the Conclusion.
(5) The references should be updated and their format standardized for enhanced consistency and accuracy.

---

### Comment · Reviewer_xVzA · 2024-08-21
**This article is very interesting and a good one**

This paper proposed and evaluated a method that uses machine learning to analyze and learn the unique features of handwritten characters. The obtained result is valuable and can be accepted if the following problems can be clarified.
(1)	In the introduction, the shortages of those relevant studies are suggested to be further summarized.
(2)	In the end of Section 1, the organization of this study is suggested to be summarized.
(3)	There exist several spelling and grammar errors. Please check carefully and further polish
(4)	The future work is missing in the Conclusion.
(5)	The references should be updated and their format standardized for enhanced consistency and accuracy.

---

### Decision · Program_Chairs · 2024-09-06

Accept (Oral)